# Lesser Tuberosity Osteotomy Healing in Stemmed and Stemless Anatomic Shoulder Arthroplasty Is Higher with a Tensionable Construct and Affected by Body Mass Index and Tobacco Use

**DOI:** 10.3390/jcm12030834

**Published:** 2023-01-20

**Authors:** Cameron Phillips, Ignacio Pasqualini, Hugo Barros, Mariano E. Menendez, Jeffrey L. Horinek, Javier Ardebol, Patrick J. Denard

**Affiliations:** 1Oregon Shoulder Institute, Medford, OR 97504, USA; 2Instituto de Hombro, Av. Rodrigo de Chávez Oe2-115, Quito 170104, Ecuador

**Keywords:** total shoulder arthroplasty, stemless, lesser tuberosity osteotomy, subscapularis healing, tensionable repair, short stem

## Abstract

Background: This study compared the healing rates of lesser tuberosity osteotomy (LTO) for anatomic total shoulder arthroplasty (TSA), repaired with either standard knot tying or a tensionable construct. Second, we evaluated LTO healing in stemmed and stemless prostheses and identified the patient characteristics associated with healing. Methods: An analysis of consecutive primary TSAs approached with an LTO performed by a single surgeon between 2016 and 2020 was conducted. In the first two years of the study period, the LTOs were repaired with four #2 polyblend sutures passed through drill tunnels and around a short press-fit stem, followed by manual knot tying. Subsequently, a tensionable construct with suture tapes (TCB) was universally adopted. The radiographic appearance of the LTO was evaluated at a minimum of six months postoperatively. Results: A total of 340 patients met the study criteria, including 168 with manual knot tying, 84 TCB repairs with a stemmed implant, and 88 TCB repairs with a stemless implant. There was no difference in the baseline demographics between the groups. The LTO healing rate of the manual knot tying group (85%) was lower than that of the stemmed (95%) and stemless (98%) TCB groups (*p* < 0.001). When directly comparing the LTO healing between the stemmed and stemless TCB groups, the differences were not significant (*p* = 0.44). Across all constructs, the body mass index (BMI) was higher in the displaced nonunion group (*p* = 0.04), with a failure rate of 9.4% for a BMI between 30 and 40, 12.5% for a BMI between 40 and 50, and 28.6% for a BMI > 50. The rate of tobacco use was higher in the displaced nonunion group (*p* = 0.037). Conclusion: A tensionable construct improves LTO healing compared to manual knot tying, irrespective of the implant type. In addition to the surgical technique, the patient factors that influence tuberosity healing include a greater BMI and tobacco use. Level of evidence: Level III, retrospective comparative study.

## 1. Introduction

Subscapularis function is critical following total shoulder arthroplasty (TSA). The three main methods to address the subscapularis during TSA include tenotomy, peel, and lesser tuberosity osteotomy (LTO) [1]. The latter technique was developed with a theoretical background of improving healing via bone-to-bone healing [2]. While there is some controversy, the LTO has generally been associated with superior biomechanical properties and healing rates [1,2,3]. The LTO poses some technical challenges, and controversy exists regarding both the repair technique and the safety of the LTO with a stemless construct.

There is still controversy about the size of the osteotomy. In most studies, the LTO is composed of small bony fragments, also known as a “wafer” or “fleck” LTO, which typically comprises 50% of the lesser tuberosity height [1,2,3]. Nevertheless, the use of larger fragments at least 100% larger than the previously reported small fragments was reported. [1,2,3]. The selection of size may vary depending on the bicipital groove depth, patient characteristics, and surgeon preference. Moreover, multiple repair techniques were proposed for the LTO, including the backpack, dual-row, and tensionable constructs [4,5]. Heckman et al. [6] found that a dual-row repair was biomechanically superior to the backpack technique. Others have suggested that the biomechanics are further improved with tensionable constructs with suture tapes (TCB) compared to a traditional suture repair [5,7]. However, currently, there is limited clinical data comparing the latter and the superiority of the repair techniques. 

With a stemmed construct, a dual-row repair is easily performed by passing sutures around or through the stem. A previous investigation demonstrated that an LTO does not compromise the fixation of a short-stem prosthesis [8]. However, one concern with the LTO is the reliability in the setting of a stemless construct [9]. While it was reported that the clinical outcomes are acceptable with an LTO in the setting of a stemless prosthesis, there is a need for data comparing LTO healing between stemmed and stemless constructs [10].

The primary purpose of this study was to compare the healing rates of an LTO performed for TSA, repaired with either standard manual knot tying or a tensionable construct. The secondary purposes were to evaluate LTO healing with the use of a tensionable construct in stemmed and stemless prostheses and to identify the patient characteristics associated with healing. Our hypothesis was that there would be no differences in the LTO healing rates based on the repair techniques or prosthesis type.

## 2. Materials and Methods

### 2.1. Study Cohort

A retrospective comparative study was performed of consecutive primary TSAs between 2016 and 2020 at a single institution. The data were reviewed from a prospective database of shoulder arthroplasties. Institutional review board approval was obtained prior to commencing the study. The inclusion criteria were a primary TSA with an intact rotator cuff utilizing an LTO approach and a minimum radiographic follow-up of six months. The exclusion criteria were revision arthroplasty, subscapularis peel or tenotomy, and incomplete follow-up. 

### 2.2. Surgical Technique

All TSAs were performed by one surgeon with 10 years of experience using a consistent technique. A deltopectoral approach was used to expose the shoulder. The bicep tendon was routinely tenodesed to the upper border of the pectoralis major tendon. An LTO was used in all cases during this period to access the glenohumeral joint. The goal of this osteotomy was to create a section that was 5 mm thick or 50% of the depth of the bicipital groove, whichever was smaller. The osteotomy did not extend into the humeral head. An anatomic cut of the humeral head was performed, respecting native retroversion and inclination. During the first three years of the study period, a short press-fit humeral prosthesis was used in all cases (Apex Univers; Arthrex, Inc., Naples, FL, USA). Thereafter, a stemless prosthesis was used (Eclipse; Arthrex, Inc., Naples, FL, USA).

From 2016 to June 2018, the surgeon repaired the LTO with four #2 sutures (Fiberwire; Arthrex, Inc., Naples, FL, USA) passed around the humeral stem and through the bone tunnels [11]. After the prosthesis was secured, each suture was tied with a 6-throw surgeon’s knot. 

In June 2018, the repair transitioned to the use of a tensionable construct [5]. In the case of the short-stem press-fit prosthesis, two holes were created in the bicipital groove, and one hole was created in the medial calcar. Then, two sets of sutures tape (FiberTape Tendon Compression Bridge; Arthrex, Inc., Naples, FL, USA) were passed from lateral to medial and around the stem. The bottom suture was passed through the inferior bicipital groove hole and through the medial calcar hole, while the superior suture set was passed through the superior bicipital groove hole and exited medially at the level of the stem (Figure 1). Each set of sutures had a pre-fashioned half-racking hitch that rested against the bicipital groove. After the prosthesis was secured, the medial suture limbs were passed through the subscapularis, then shuttled through the half-racking hitches. (Figure 2 and Figure 3). The slack was manually removed, and then a single-half hitch was applied. The half-racking hitch construct provided the ability to secure the sutures provisionally and was resistant to suture slippage [12]. Next, the osteotomy was compressed with a dedicated tensioner (Arthrex, Inc., Naples, FL, USA). The free limbs were threaded through a tensioner, and the limbs were wrapped onto a mobile thumbpad of a tensioning device. By turning the handle of the device, the sutures were pulled away from the half-racking hitch, removing all slack from the construct (Figure 4). Tensioning was performed under direct visualization and tactile feedback. Finally, the construct was locked with two half hitches (Figure 5).

In the case of the stemless prosthesis, the same tensionable construct was used, except that the sutures were passed through the extraction slots of the implant. In all repair techniques, the repair was performed with the arm in 20 to 30° of external rotation and neutral extension. No anchors were used in any of the repairs. Additionally, in all techniques, the rotator interval was closed with three to four #2 sutures placed in a simple suture configuration.

Postoperative rehabilitation was standardized in all cases [13]. The patients wore a sling for four weeks postoperatively with hand, wrist, elbow, and neck motion only. At four weeks, the sling was removed, and a passive range of motion was initiated. Strengthening was allowed at two months postoperation, followed by a gradual return to normal activities.

### 2.3. Radiographic Evaluation

Radiographic anteroposterior, axillary lateral, and scapular Y views were obtained preoperatively and at the final follow-up visit. The radiographic appearance of the LTO was evaluated at a minimum of six months postoperation on plain radiographs and categorized as a bony union, nondisplaced nonunion, or displaced nonunion [14]. For analysis purposes, the bony and nondisplaced unions were further classified as healed. The radiographs were independently evaluated by two orthopedic surgeons (PJD and HB). Any cases of disagreement were rectified by re-review and mutual agreement.

### 2.4. Statistical Analysis

The statistical analyses were performed by a trained statistician. The continuous variables are presented as the mean ± standard deviation (SD) or the median and interquartile range (IQR) according to the distribution, and the categorical variables are presented as the absolute and relative frequencies. A one-way ANOVA or the Kruskal-Wallis test was used to compare the continuous variables between the groups according to their assumptions. The chi-square test or the Fisher exact test was used to compare the categorical variables between the groups according to their assumptions. To evaluate our hypothesis that there would be no differences in the healing rate based on the repair technique or prosthesis type, a chi-square test was used. *p* values under 0.05 were considered statistically significant. The statistical analyses were performed using SPSS version 25 (IBM Corp, Armonk, NY, USA).

## 3. Results

### 3.1. Study Population 

A total of 359 TSAs were performed during the study period. A total of 19 pateints were lost at the follow-up, leaving 340 (94.7%) patients that met the study criteria, including 168 with manual knot tying, 84 TCB repairs with a stemmed implant, and 88 TCB repairs with a stemless implant. There was no difference in the baseline demographics between the groups (Table 1).

### 3.2. Healing Based on the Repair Technique

The overall LTO healing rate was 90.5 %. LTO healing was significantly different between the groups, with the highest rate observed in the stemless TCB group (98%) (Figure 6), followed by the stemmed TCB group (95%) (Figure 7), and then the manual knot tying group (85%) (*p* < 0.001) (Table 2).

### 3.3. Healing Based on the Patient Characteristics

Across all constructs, there were no differences in healing based on age or sex. However, the preoperative body mass index (BMI) was higher in the displaced nonunion group (*p* = 0.04), with a failure rate of 9.4% for a BMI between 30 and 40, 12.5% for a BMI between 40 and 50, and 28.6% for a BMI > 50 (Figure 8). The rate of preoperative tobacco use was also higher in the displaced nonunion group (*p* = 0.037) (Table 3).

## 4. Discussion

The primary finding of the current study was that while the overall LTO healing rate was high, significant differences were found between the repair techniques, with higher rates observed with a tensionable construct compared to a traditional knot tying construct. Secondly, the rate of LTO healing using a tensionable construct was similarly high whether a stemmed or stemless implant was used. Finally, an increased BMI and tobacco use were associated with tuberosity nonunion. These findings have important clinical implications for LTO usage and repair techniques in the setting of TSA. 

Regardless of the technique, subscapularis healing is well-recognized to be of high importance following TSA. Following reports of poor healing with tenotomy, the LTO was developed as an alternative approach to achieve bone-to-bone healing. In a recent systematic review, Del Core et al. [2] compared healing rates based on the subscapularis approach. In eight studies reporting on LTO healing at a minimum of three months postoperation, the weighted mean healing rate was 98.9%. In the four studies that assessed peel and tenotomy, the healing rates were comparatively lower (74.1% and 75.9%, respectively; *p* = 0.002). These results were also consistent with previous reports [1]. However, the LTO healing rates are lower in other series. For instance, Levy et al. [15] studied radiographic LTO healing following TSA using a press-fit humeral component, showing a rate of 75.7%. Our overall 90.5% rate of healing is within the range of previous literature reporting on LTO healing. Additionally, we observed that preoperative patient factors associated with a displaced LTO nonunion included an increased BMI and tobacco use. Previous studies have linked these factors with worse outcomes following TSA but have not specifically linked BMI or tobacco use to the failure of LTO healing [14,16,17,18]. Small et al. [14] suggested that tobacco use may influence LTO nonunion but did not find statistical differences. Nonetheless, they suggested that surgeons should counsel against tobacco use prior to TSA. Our findings reiterate this importance. Additionally, the dramatic decrease in LTO healing we observed with an increased BMI, particularly over 50, where the nonunion rate was 29%, is important to note. This information could be used for preoperative counseling, TSA avoidance, or even RSA consideration in these patients [19].

A variety of repair techniques were proposed for LTOs [4,5]. Although no standard repair technique exists, a combination of tension and compression sutures is the best with biomechanical properties [4]. Single-row and dual-row repair techniques were also described [6]. Moreover, sutures passed around the stem have demonstrated superior biomechanical outcomes than using just tension sutures [4,11,20]. Adding a simplified technique using a tensionable construct was associated with improved biomechanical properties than a traditional LTO repair [5]. Although the latter technique showed favorable results, there is a paucity of literature on LTO healing compared to other repair techniques or between different prostheses. Our results further suggest that using a tensionable construct may increase LTO healing compared to traditional repair techniques independent of the type of prosthesis; indeed, we found that both groups using a TCB showed significantly higher healing rates than manual knot tying (stemless TCB group 98%, stemmed TCB group 95%, and manual knot tying 85%).

Over time, the humeral prosthesis length decreased in an effort to maximize bone stock, facilitate revision, and avoid stress shielding [21,22]. Excellent results were published following TSA using short-stemmed or stemless prostheses [9,15,23,24,25,26]. While there is debate on performing an LTO in these kinds of prostheses as there is a possible risk of compromising proximal fixation, Griffin et al. [8] recently reported that the LTO was safe to be performed in short-stem constructs [9]. However, the literature has not been as clear for a stemless prosthesis. In a multicenter study, Aibinder et al. [10] evaluated subscapularis management techniques in stemless TSA, reporting that of the 55 patients in the LTO group, none were found to have displacement on axillary radiographs [10]. Our study found that high LTO healing rates could be obtained by performing TSA with a stemless construct. Moreover, no differences were found when comparing the healing rates using tensionable constructs between stemmed and stemless prostheses. Therefore, it appears that performing an LTO on these types of prostheses is safe and yields excellent healing results. 

This study is not without limitations. First, as a retrospective study, it has all the limitations inherent to this kind of study. Second, we did not perform a power analysis before testing healing among the types of prostheses and repair techniques. While differences were observed between the tensionable and hand-tying techniques, the study may not have been powerful enough to detect the differences between the stemmed and stemless tensionable repair groups. However, the differences between the latter were small and thus not likely clinically relevant. Third, there was no control group using other subscapularis management techniques. Fourth, we did not evaluate the influence of the impact of the LTO on the prosthesis stability. While we did not observe any case of stemless loosening in this series, this was not specifically evaluated as it was not the purpose of this investigation. Fifth, we did not evaluate other factors that could be associated with healing, such as humeral head sizing and the quality of reduction. Finally, we did not evaluate the subscapularis function through physical examination tests or strength measurements, and we did not evaluate the overall functional outcomes, as the purpose of this study was only to assess radiographic LTO healing, and this was reported by previous authors [2,13,15,27,28].

## 5. Conclusions

A tensionable construct improves LTO healing compared to manual knot tying, irrespective of the implant type. This study adds to the growing evidence that the LTO could be safely performed with stemless implants and achieve high union rates. In addition to the surgical technique, the patient factors that influence tuberosity healing include a greater BMI and tobacco use.

## Figures and Tables

**Figure 1 jcm-12-00834-f001:**
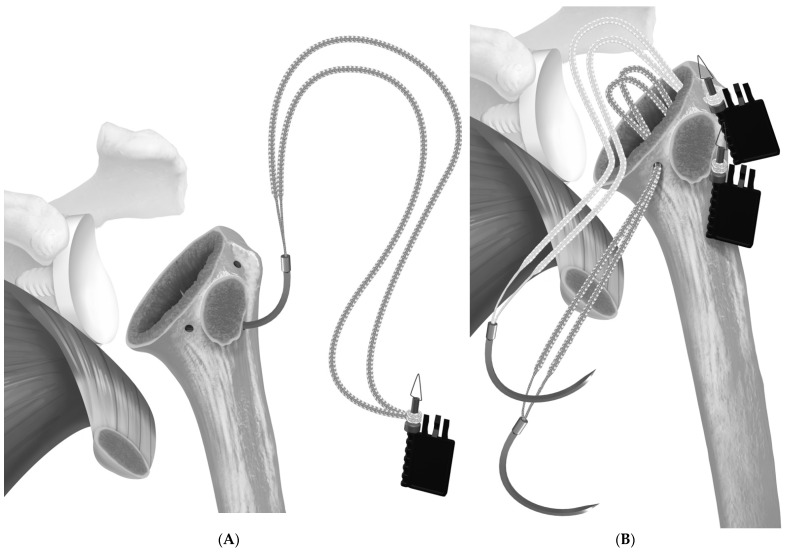
(**A**) suture tape pre-fashioned with a half-racking suture on the end passed from lateral to medial through the inferior two holes, and (**B**) a separate suture is passed through the superior hole. Reproduced with permission from [5].

**Figure 2 jcm-12-00834-f002:**
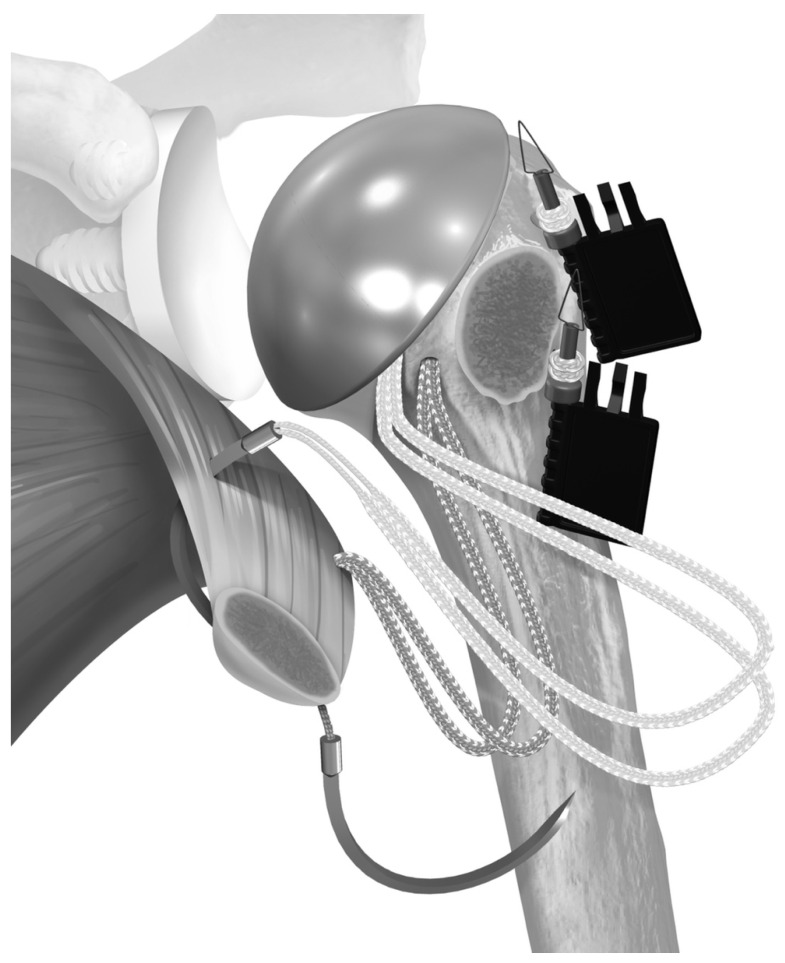
The stem is placed so that the sutures pass around the prosthesis. Then, the sutures are passed through the subscapularis tendon. Reproduced with permission from [5].

**Figure 3 jcm-12-00834-f003:**
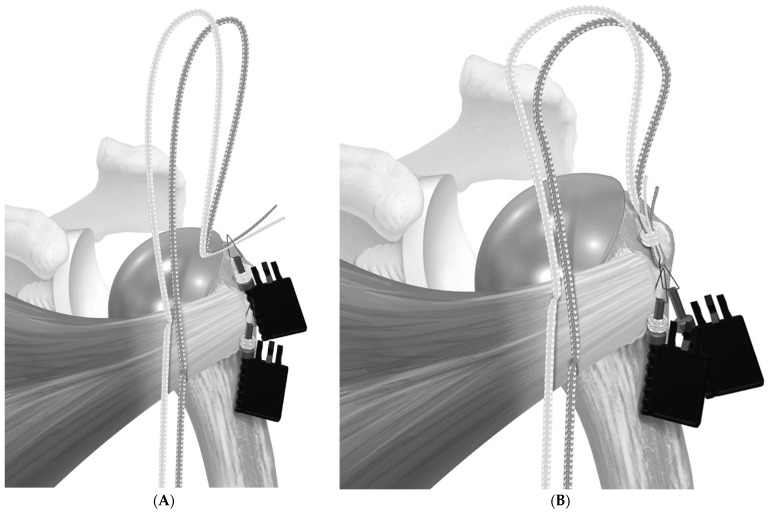
One suture limb from each pair was selected (**A**) and passed through the pre-fashioned half-racking suture. (**B**) Reproduced with permission from [5].

**Figure 4 jcm-12-00834-f004:**
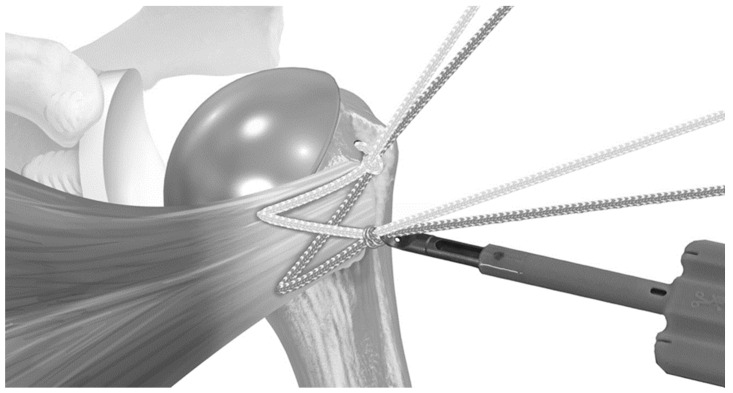
The suture limbs passed through the half-racking suture were tensioned with a dedicated tensioning device under visual inspection. Reproduced with permission from [5].

**Figure 5 jcm-12-00834-f005:**
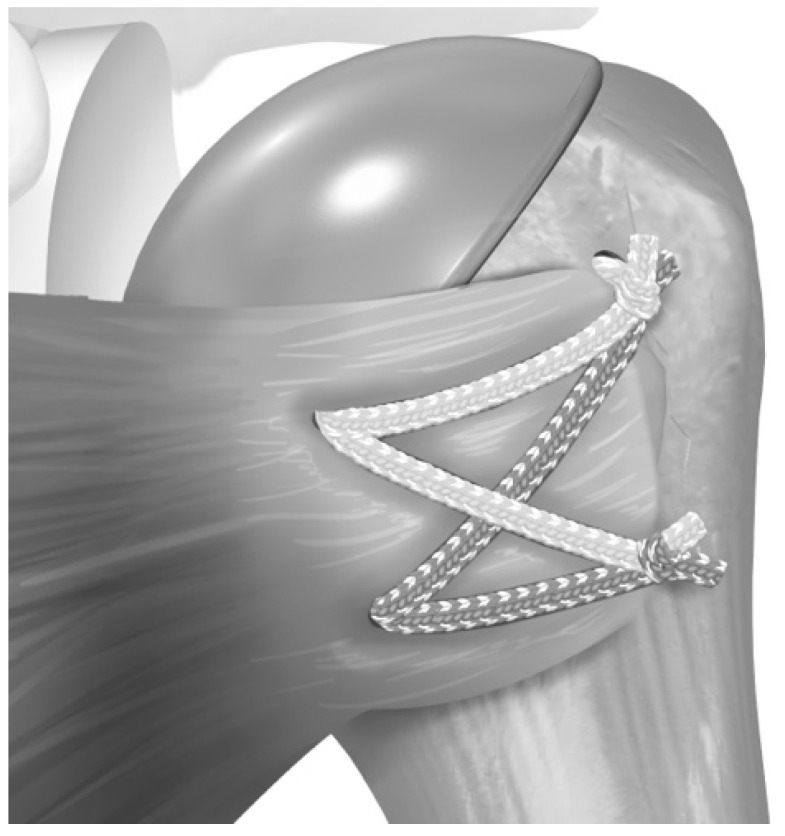
Final tenisonable repair construct. Reproduced with permission from [5].

**Figure 6 jcm-12-00834-f006:**
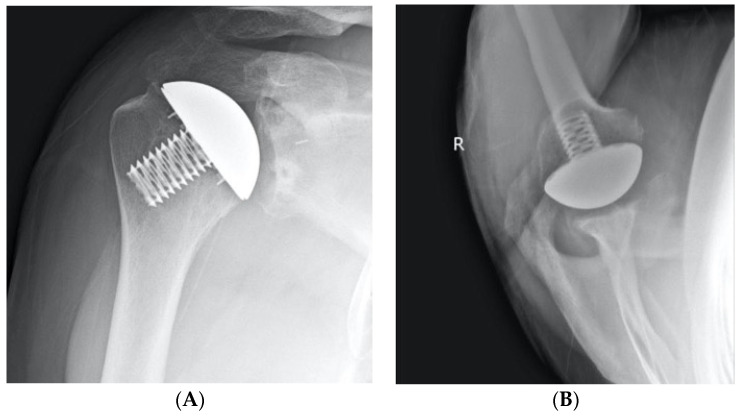
Anteroposterior (**A**) and axillary (**B**) radiographs of a left shoulder demonstrate a healed lesser tuberosity osteotomy following stemless total shoulder arthroplasty.

**Figure 7 jcm-12-00834-f007:**
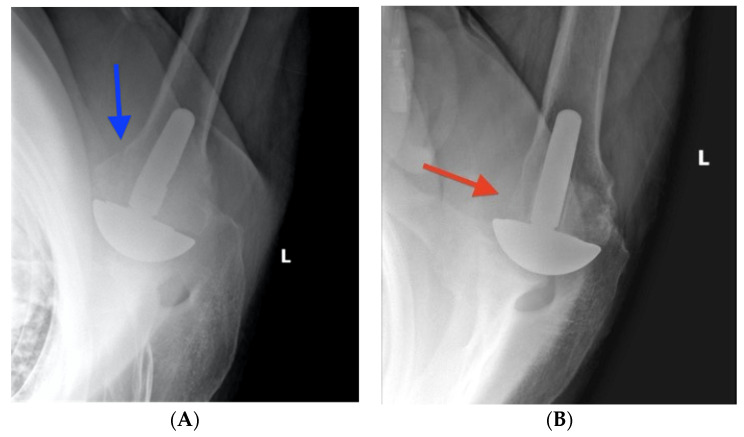
(**A**) Initial postoperative axillary radiograph in a left shoulder demonstrates an intact lesser tuberosity osteotomy (blue arrow) following a short-stem total shoulder arthroplasty. (**B**) Six-month postoperative radiographs show the absence of the osteotomy (red arrow).

**Figure 8 jcm-12-00834-f008:**
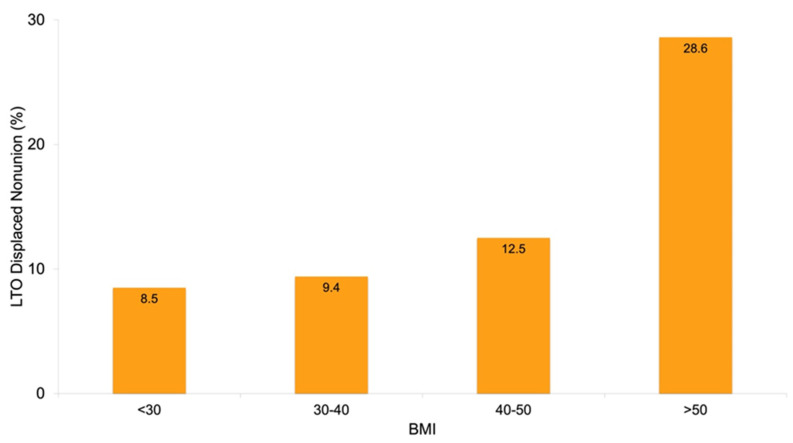
LTO healing based on BMI.

**Table 1 jcm-12-00834-t001:** Characteristics of the study population.

Parameter	All Patients	LTO Repair Technique			
		Knot Tying with a Short Stem	TCB with a Short Stem	TCB with Stemless	*p*
Total †	340 (100)	168 (49.4)	84 (24.7)	88 (25.9)	
Age * (yr)	68.6 ± 8.2	68.2 ± 8.0	69.7 ± 8.6	68.2 ± 8.1	0.326
Sex †					
Female	156 (45.9)	73 (43.5)	47 (56.0)	36 (40.9)	0.095
Male	184 (54.1)	95 (56.5)	37 (44.0)	52 (59.1)	
BMI *	30.8 ± 6.9	31.0 ± 7.1	29.7 ± 6.8	31.4 ± 6.5	0.297
Tobacco use †	10 (2.9)	6 (3.6)	0	4 (4.6)	0.165
Implant type †					
Stemless	88 (25.9)	0	0	88 (100)	
Short-stem	252 (74.1)	168 (100)	84 (100)	0	

BMI = body mass index; TCB = tensionable construct. * The values are represented as the mean and the standard deviation. † The values are represented as the number of patients, with the percentage in parentheses.

**Table 2 jcm-12-00834-t002:** Lesser Tuberosity Osteotomy Healing Stratified by Repair Technique.

Parameter	LTO Repair Technique			
	Knot Tying with a Short Stem (n = 168)	TCB with a Short Stem (n = 84)	TCB with Stemless (n = 88)	*p*
LTO †				
Displaced nonunion	26 (15.5)	4 (4.8)	2 (2.3)	<0.001 *
Healed or nondisplaced nonunion	142 (84.5)	80 (95.2)	86 (97.7)	
LTO †				
Displaced nonunion	26 (15.5)	4 (4.8)	2 (2.3)	<0.001
Nondisplaced nonunion	26 (15.5)	24 (28.6)	16 (18.2)	
Healed	116 (69.0)	56 (66.7)	70 (79.5)	

TCB = tendon compression bridge. † The values are represented as the number of patients, with the percentage in parentheses. * *p* = 0.44 when excluding the No. 2 Fiberwire group.

**Table 3 jcm-12-00834-t003:** Lesser Tuberosity Osteotomy Healing Stratified by Patient Characteristics.

Parameter	LTO			
	Healed (n = 242)	Nondisplaced Nonunion (n = 66)	Displaced Nonunion (n = 32)	*p*
Age * (yr)	69.0 ± 7.9	67.1 ± 8.9	68.2 ± 9.1	0.233
Sex †				
Female	117 (48.3)	28 (42.4)	11 (34.4)	0.270
Male	125 (51.7)	38 (57.6)	21 (65.6)	
BMI *	30.8 ± 6.7	29.2 ± 6.7	33.1 ± 8.2	0.043
Tobacco use †	7 (2.9)	0	3 (9.4)	0.037

* The values are represented as the mean and the standard deviation. † The values are provided as the number of patients, with the percentage in parentheses.

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
