# Peer review of "Lesser Tuberosity Osteotomy Healing in Stemmed and Stemless Anatomic Shoulder Arthroplasty Is Higher with a Tensionable Construct and Affected by Body Mass Index and Tobacco Use"

_jcm, 2023, doi:10.3390/jcm12030834_

Round 1
Reviewer 1 Report
This study aims to compare the healing rates of lesser tuberosity osteotomy (LTO) for anatomic total shoulder arthroplasty (TSA), repaired with either standard knot tying or a tensionable construct and to evaluate LTO healing in stemmed and stemless prostheses, and identified patient characteristics associated with healing.
This is an interesting study around the healing of the lesser tuberosity with a specific analysis of two methods of suture and stem or stemless humeral implant.
One point is that the authors justify the study by the important clinical implication of the healing of the lesser tuberosity (dislocation, internal rotation...) but the reader would expect an evaluation of the clinical consequences of the “non-healed” lesser tuberosity. Same between non-union/non-displaced LT or non-union/displaced LT. With such information, the study would have a more important impact and would not look like a “database study”.
We guess that over 340 TSA, some complications might occur.
The X-ray follow-up is never given.
Figures are missing and the reader must be sure about what is a healed, displaced or non-healed LT.
Specific comments:
Retrospective design must be clearly written.
As a retrospective study, the population with included patients must be described in the Methods section
Methods:
Figures including X-Rays must be added to show one or some cases of LTO non-union
X-Ray evaluation: do the authors have any data about the interobserver analysis discrepancies?
Results
Population must be described (Line 149 to 160) in the Methods section
The last follow-up (X-rays evaluation especially) must be specified in the Table 1
Table “p” for line “Stemless” seems difficult to interpretate or not of interest
Table 2 and 3: the number of patients by column‘item must be specified
Discussion
Pertinent
Line 129: first time we see the word “retrospective”
Author Response
"Please see the attachment.

Reviewer 2 Report
This manuscript entitled "Lesser Tuberosity Osteotomy Healing in Stemmed and Stemless Anatomic Shoulder Arthroplasty is Higher with a Tensionable Construct and Affected by Body Mass Index and Tobacco Use" focused an interesting topic regarding different repair methods for lesser tuberosity osteotomy. The manuscript is well organized and very readable. The tensionable construct with suture tapes is a simple method that shows a high healing rate and is considered to be an excellent method. However, readers would also like to know if the healing rate affects clinical outcome. Therefore, if possible, I would like you to include whether the repair method affects the postoperative outcome.
Author Response
"Please see the attachment

Round 2
Author Response
Dear reviewer,
Thank you for your review of our article entitled: “Lesser Tuberosity Osteotomy Healing in Stemmed and Stemless Anatomic Shoulder Arthroplasty is Higher with a Tensionable Construct and Affected by Body Mass Index and Tobacco Use”
Below is our response to the review and manuscript edits. We hope you find these satisfactory.
Regarding your report form suggestion for improving the study design, we added a limitation about the lack of a power analysis as the sample size was limited by the study period. Lines 255-259
Finally, regarding the improvement of the result section, we respecfully disagree with this suggestion as we believe our results are clearly reported.